# Spectral and Geometrical Guidelines for Low-Concentration Oil-in-Seawater Emulsion Detection Based on Monte Carlo Modeling

**DOI:** 10.3390/s25175267

**Published:** 2025-08-24

**Authors:** Barbara Lednicka, Zbigniew Otremba

**Affiliations:** Department of Physics, Gdynia Maritime University, 81-225 Gdynia, Poland; z.otremba@wm.umg.edu.pl

**Keywords:** seawater, oil sensor, dispersed oil, radiance field, scalar irradiance, Monte Carlo simulation

## Abstract

This paper is a result of the search for design assumptions for a sensor to detect oil dispersed in the sea waters (oil-in-water emulsions). Our approach is based on analyzing changes in the underwater solar radiance (L) field caused by the presence of oil droplets in the water column. This method would enable the sensor to respond to the presence of oil contaminants dispersed in the surrounding environment, even if they are not located directly at the measurement point. This study draws on both literature sources and the results of current numerical modeling of the spread of solar light in the water column to account for both downward and upward irradiance (Es). The core principle of the analysis involves simulating the paths of a large number of virtual solar photons in a seawater model defined by spatially distributed Inherent Optical Properties (IOPs). The IOPs data were taken from the literature and pertain to the waters of the southern Baltic Sea. The optical properties of the oil used in the model correspond to crude oil extracted from the Baltic shelf. The obtained results were compared with previously published spectral analyses of an analogous polluted sea model, considering vertical downward radiance, vertical upward radiance, and downward and upward irradiance. It was found that the optimal wavelength ratio of 555/412, identified for these quantities, is also applicable to scalar irradiance. The findings indicate that the most effective way to determine this index is by measuring it using a sensor with its window oriented in the direction of upward-traveling light.

## 1. Introduction

Traditional human activity in maritime areas, i.e., transport techniques (navigation), fishing, military, coastal development (ports, coastal protection, tourist facilities), and offshore development (bridges, dams, tunnels), is increasingly supplemented by various large-scale technical marine structures—industrial constructions. Additionally, developing marine oil and gas resources requires a wide range of technical infrastructure [1]. High-power electricity transmission systems, oil and natural gas pipelines, oil industry exploration, production and technological platforms, electricity generators (conversion of wave energy, sea currents, and wind), and offshore trans-shipment ports are examples of technical structures permanently located at sea. Currently, there are more than 10,000 offshore oil and gas platforms around the world, with many in operation for over 15 years [2]. Between 1970 and 2023, there were 471 major oil spill incidents, each involving a spill of more than 700 tonnes [3]. Moreover, offshore oil extraction contributes to roughly 25% of the world’s total oil production [4]. Therefore, there is an urgent need for a complete optical profile for oil detection and tracking in marine areas [5].

Industrial installations emit pollutants in the form of admixtures of chemical compounds (e.g., hydrocarbons) and energies foreign to the marine environment (static electric and magnetic fields, electromagnetic fields, acoustic noise). Some of these structures are potential emitters of harmful substances that enter the sea in dispersed form. These are mainly hydrocarbons, as residues of crude oil, lubricating oils, and fuels. Such substances pass into suspension due to the action of surface waves and due to the use of special substances (dispersants) that aid dispersion. The tracking and monitoring of oil-in-water emulsions in seawater are crucial for mitigating the negative environmental and economic impacts of oil spills. In the case of removal of effects of oil spills, different methods are used (i.e., anti-spill floating booms combined with skimmers, chemical dispersants, or even in situ burning) [6,7,8,9,10]. However, once oil builds up in marine environments, it becomes challenging to eliminate and may require decades for the coastal ecosystem to fully recover [11,12,13].

In this paper an interdisciplinary approach is proposed, using both modeling and empirically measured values, to create tools for tracking anthropogenic impact (oil pollution in this case) on the marine environment. Designing remote measuring systems that respond to changes in solar radiance caused by anthropogenic pollution, while also capturing these changes at the greatest possible depth, is currently one of the challenges [14,15].

In this study, we focus on digitally modeling an underwater optical sensor designed specifically to measure the presence and concentration of oil pollutants throughout the water column, not just at the surface. Our goal is to support the development of a monitoring system capable of detecting dispersed oil at different depths of the sea.

In areas of the sea exposed to the presence of oil, a key element of the underwater remote monitoring system would be a meter that generates a signal informing about changes in the water caused by the presence of oil substances. It is a valid assumption that the presence of a cloud of oil droplets (oil-in-water) causes a modification of the radiance field. It can be expected that such an optical trace of oil also occurs up to a certain distance outside the cloud of oil droplets. Radiance in the water column is a function of several variables, primarily the spectral and angular distribution of solar light falling on the sea surface, the state of the sea surface, depth, and the Inherent Optical Properties (IOPs) of seawater, which consist of the optical properties of solid and dissolved components in water. Changes in the radiance field in the sea can be caused not only by oil emulsion but also by other factors, such as bubbles, resuspended bottom sediment, and planktonic blooms. The possibility of detecting oil in seawater results from its characteristic spectral optical properties, which differ from the optical properties of seawater. The spectral dependencies of the absorption coefficient, scattering coefficient, and angular scattering function are important here. The extent to which the indices change, representing the ratio of radiance at a specific wavelength to radiance at other wavelengths, was analyzed. It was shown that for the vertical direction of radiance, the index 555/412 is optimal, both for the radiance meter directed upwards [16] and downwards [17].

The negative feature of the radiance meter is that due to the narrow solid angle from which the light is received, it is required to work in good lighting conditions. In connection with this, the possibility of using a vector irradiance meter, which collects light from the entire hemisphere (upper or lower), was investigated [18].

Vector irradiance is defined by the angular distribution of radiance *L* (θ,φ). In the case of downward vector irradiance, its definition [19] is given by the expression(1)$$E_{v\_d}\left(\theta ,\varphi \right)=\int_{0}^{2\pi }\int_{0}^{\frac{\pi }{2}}L\left(\theta ,\varphi \right)\sin\theta \cos\theta d\theta \;d\varphi ,$$
where *θ* and φ are angular coordinates called directional angles, specifically, zenith and azimuth, while in the case of upward vector irradiance, its definition [19] is given by the expression(2)$$E_{v\_u}\left(\theta ,\varphi \right)=\int_{0}^{2\pi }\int_{\frac{\pi }{2}}^{\pi }L\left(\theta ,\varphi \right)\sin\theta \cos\theta d\theta \;d\varphi .$$

Vector irradiance takes into account vertical radiation to a greater extent than radiation incident from the side. Scalar irradiance, on the other hand, takes into account radiance from all directions equally, so a scalar irradiance meter receives more light than a vector irradiance meter. Scalar irradiance, similar to vector irradiance, is defined by the angular distribution of radiance. In the case of downward scalar irradiance, its definition [19] is given by the expression(3)$$E_{s\_d}\left(\theta ,\varphi \right)=\int_{0}^{2\pi }\int_{0}^{\frac{\pi }{2}}L\left(\theta ,\varphi \right)\sin\theta d\theta \;d\varphi ,$$
while in the case of upward scalar irradiance, its definition [19] is given by the expression(4)$$E_{s\_u}\left(\theta ,\varphi \right)=\int_{0}^{2\pi }\int_{\frac{\pi }{2}}^{\pi }L\left(\theta ,\varphi \right)\sin\theta d\theta \;d\varphi .$$

In this work, it was checked whether in the case of downward and upward scalar irradiance the index 555/412 is also the most favorable index, as in the case of downward and upward radiance as well as in the case of downward and upward vector irradiance. Furthermore, it was identified which of these six cases would be the most favorable in terms of design principles of a possible remote sensing system for underwater oil emulsion clouds.

## 2. Materials and Methods

The trace of a large number of virtual solar photons—the Monte Carlo method (MC)—was used to model the upward and downward scalar irradiance. Figure 1 presents a block diagram explaining the steps involved in obtaining scalar irradiance below the sea surface. The virtual model of the sea area was based on a few assumptions: the water surface was virtually wavy as a result of a wind speed of 5 m/s (based on Cox and Munk [20]); there was a cloudless sky; and the angle of incidence of rays directly from the sun was 30°. The computational algorithm counted virtual photons in seawater polluted with both oil-in-water emulsion and oil-free seawater. The model studies were based on in situ measurements of scattering and absorption coefficients (after Sagan [21], Table 1) for the southern Baltic Sea and on Petrobaltic-type crude oil extracted from this area (Table 2). The virtual receivers were placed at various depths. In the optical model of an oil-polluted water layer of 30 m, an oil droplet concentration of 10 ppm was assumed. As a result, eight wavelengths spanning almost the entire spectral range were chosen for this study, from 412 nm to 676 nm. Most of the ocean color algorithms are based on the parameterization of Apparent Optical Properties (AOPs) (e.g., the diffuse attenuation coefficients for downward irradiance or remote sensing reflectance) as functions of IOPs and water constituents [22]. IOPs play a significant role in determining the light field in seawater. In recent decades, numerous hydrologic optical instruments (e.g., AC9 or ACS from WET Labs Inc) have been employed in the field to obtain in situ values of IOPs [23,24,25]. Therefore, in this study, wavelengths were selected according to the measurement capabilities of AC9.

## 3. Results

Figure 2 presents the results of the staged simulations, specifically the angular distributions of downward and upward radiance for both oil-free and oil-contaminated seawater at a selected wavelength (555 nm) and depths (1 m and 5 m). The radiance distributions were obtained by simulating the trajectories of a large number of virtual photons (one hundred million) within both clean and oil-polluted marine environments. Significant differences are observed between the two cases. Based on these distributions, both scalar and vector irradiance values were calculated.

Figure 3 shows a cross-section of downward radiance for oil-polluted seawater and for free-of-oil seawater for greater depths like in Figure 2 (1, 2, 3, 4, and 5 m for oil-polluted seawater and 1, 2, 3, 4, 5, 7, and 10 m for oil-free seawater) for the chosen wavelength of 555 nm. In the case of oil-polluted seawater at a depth of 5 m, negligible radiance was recorded, whereas in the case of free-of-oil seawater, the irradiance was recorded even at a depth of 10 m. However, in both cases, these values decrease with depth, and there is a visible flattening. However, there is no evidence that in the real case such changes would be caused only by oil and not by natural components of seawater. Therefore, simulations were performed for different wavelengths.

Based on the directional radiance distributions (illustrated in Figure 2 and Figure 3), the scalar irradiance values were calculated. The diversity of scalar irradiance at different depths and at different wavelengths is shown in the logarithmic scale in Figure 4 and Figure 5. Figure 4 shows differences between downward and upward scalar irradiance for oil-free seawater, while Figure 5 shows the same but in oil-polluted water. The linearity of the graphs is maintained only up to a depth of 5 m. At greater depths the slope of the graph changes due to the change in the IOPs of the seawater used in the sea model.

Figure 4 and Figure 5 show that the irradiance values decrease rapidly, especially for wavelengths 412, 440, and 676 nm. The photons arriving from down were in trace quantities at a depth of 10 m, especially for oil-polluted water (Figure 5). This may be due to the fact that for Baltic Sea waters, the minimum absorption occurs in the green, yellow, and orange wavelengths, i.e., in the range from approximately 460 to 600 nm. However, shorter wavelengths are absorbed by Chromophoric Dissolved Organic Matter (CDOM) and longer wavelengths by water molecules [19].

The spectral index values for all 28 combinations of wavelengths for downward and upwelling scalar irradiance were determined. The spectral indexes for seawater polluted with an oil-in-water emulsion and seawater free of oil for irradiance readings for four depths of 1 m, 2 m, 5 m, and 10 m were defined as the ratio of the upwelling scalar irradiance for the longer wavelengths to the upwelling scalar irradiance for the shorter wavelengths and as the ratio of the downward scalar irradiance for the longer wavelengths to the downward scalar irradiance for the shorter wavelengths. The highest value of the index for oil-free water and dispersed oil is recorded for the wavelength combination 555/412 (Figure 6). Moreover, Figure 6 also shows the differences between the values of the spectral index for oil-polluted seawater and oil-free seawater for scalar irradiance readings. The values of the spectral index increase with decreasing depth, and these indicators are ten times higher (for depths of 5 m and 10 m) for oil-polluted than oil-free waters.

Figure 6 consists of two columns of polar plots that analyze differences in the spectral index values between oil-polluted and oil-free seawater under chosen depths (1 m, 2 m, 5 m, 10 m). Specifically, it focuses on scalar irradiance: downward scalar irradiance (light coming from the above-left column) and upward scalar irradiance (light coming from the below-right column). The radial axis of Figure 6 represents the magnitude of the difference in spectral index between polluted and oil-free water. Whereas the angular axis represents various spectral index combinations, marked by wavelength ratios *λ_i_/λ_i_*, where *λ_i_* = 412 nm, 440 nm, 488 nm, 510 nm, 532 nm, 555 nm, 650 nm, and 676 nm (e.g., 532/440, 488/412, etc.).

Table 3 and Table 4 show that for a depth of 1 m and for downward scalar irradiance, very small differences are observed (1.5 maximum). On the other hand, for upwelling scalar irradiance, slightly more pronounced changes are observed (8.4 maximum). For 2 m downward, scalar irradiance shows moderate differences (maximum 10.1). However, upwelling scalar irradiance is characterized by a significant increase (maximum 47.1). Oil pollution starts to strongly affect upwelling scalar irradiance, especially around the green–blue region (e.g., 555/412). At a depth of 5 m, a large increase in the spectral index for downward scalar irradiance readings is observed (maximum 1103), showing that oil pollution has cumulative effects on how light propagates downward as it travels deeper. For downward scalar irradiance readings for the same depth, a huge spike is observed (maximum 4137). Finally, for a depth of 10 m and for downward scalar irradiance, a very large difference is observed (maximum 194,865). For upward scalar irradiance, the maximal difference amounts to 189,257, matching the downward trend, meaning the spectral signature of oil becomes very strong and distinct at such depth.

It was confirmed that the spectral index of 555/412 is the most favorable combination of two wavelengths for the underwater detection of oil-in-water emulsions. Based on a comparison of the present and previous studies, the present results provide strong evidence that a scalar irradiance should be chosen instead of a radiance or vector irradiance sensor for detection of oil-in-water emulsions (Figure 7). This is also a highly advantageous result from a technical perspective, as a downward-facing sensor is less likely to accumulate various types of contaminants, such as organic and mineral deposits.

This study evidenced that the sensitivity of the potential oil detector increases during its submersion. At the same time, it is worth noting that the amount of light decreases with depth, which could prevent the detector from working. However, while radiance meters become virtually useless at low light levels, scalar irradiance sensors can still operate effectively, providing more efficient data acquisition in difficult lighting conditions. This is due to the cumulative optical effects that oil has on light as solar photons travel longer in water. The numbers in Table 3 and Table 4 are the result of the simulation of the migration of a large number of virtual photons in the ocean depths. Each photon is subject to the probability of scattering or absorption (‘death’) at each section of its path. These probabilities are a strict consequence of the absorption and the scattering coefficients. In this article, the values of these coefficients that are realistic for the southern region of the Baltic Sea were assumed. Therefore, if we generalize the conclusions from the obtained results, they should be treated as qualitative but certainly reliable in relation to the guidelines for the design of a system for monitoring the purity of waters at risk of contamination with oils dispersed in water. It is possible that our findings could be useful for developing or refining algorithms in ocean color remote sensing, especially in detecting oil spills from satellite or aerial platforms. Moreover, in the future, integration of values obtained on the basis of scalar irradiance sensors proposed in this work and multi-parameter algorithms for remote sensing reflectance, absorption, or backscattering mentioned in [26,27,28,29] can provide enhanced tools for tracking and monitoring oil-in-water emulsions, particularly in optically complex water areas like those in the southern Baltic Sea.

This study builds upon previous research in the field of marine remote sensing by incorporating an additional and often overlooked variable: the total amount of light energy available in all directions at a given point underwater, also known as scalar irradiance. While earlier studies primarily focused on directional light measurements such as radiance or vector irradiance, this work emphasizes the importance of capturing the omnidirectional nature of light as it interacts with the aquatic environment.

Importantly, the findings demonstrate that oil pollution can be detected even when the oil is present in the form of highly diluted emulsions. This highlights a significant advancement in detection capability, as it opens the door to identifying subtle or early-stage pollution events that might otherwise go unnoticed until they manifest more severely.

The omnidirectional sensitivity of the detector is particularly beneficial in deeper waters, where light levels are naturally reduced and directional measurements using radiance meters become less reliable. The ability to detect subtle differences in spectral indices at greater depths extends the range of detection instruments based on the scalar irradiance meter principle, making them more suitable for long-term environmental monitoring in marine environments. For example, while radiance meters become virtually useless at low light levels, scalar irradiance sensors can still operate effectively, providing more consistent data acquisition in difficult lighting conditions.

Another key observation is the depth-dependent behavior of detector sensitivity. The results show a notable increase in the sensitivity of the spectral index to oil presence as the sensor is submerged deeper. This is likely due to the cumulative optical effects that oil has on light as it travels through water. However, this increased sensitivity is limited by a concurrent decrease in the total available light energy at greater depths, which may impose practical limits on the sensor’s effectiveness if ambient light falls below a functional threshold.

The findings suggest that a scalar irradiance meter would serve as the most efficient type of light detector for this purpose. Unlike vector irradiance sensors, which are directionally dependent, a scalar irradiance meter collects light from all upward directions within the water column, making it particularly suited for capturing diffuse light. This is critical in marine environments, where suspended matter and varying light conditions can significantly affect detection accuracy.

The simulations and analysis were conducted with a model of seawater exhibiting relatively high turbidity, characteristic of the Baltic Sea. In such environments, light attenuation is more pronounced due to the presence of suspended particulate matter, CDOM, and other optically active substances. Despite these challenges, the results showed that it should still be possible to detect oil pollution by optimizing the detection parameters, such as wavelength selection and sensor orientation.

It is worth noting that in marine areas with higher water transparency, such as open ocean waters or tropical seas, the detection of oil dispersed in the water column is expected to be more effective. This is because lower turbidity allows light to travel further, enhancing the detector’s ability to distinguish oil from the surrounding water. Therefore, while the system modeled here is optimized for the relatively challenging optical conditions of the Baltic Sea, it holds even greater promise for application in clearer waters.

## 4. Conclusions

This research focused on identifying optimal structural and functional parameters for an optical sensor capable of detecting environmentally hazardous oil substances, in this case, oil dispersed in seawater (oil-in-water emulsion). Through numerical simulation, this study confirms that employing a measurement system based on the detection of solar light intensity at two specific wavelengths, approximately 555 nm and approximately 412 nm, is optimal. The findings presented in this work suggest that a scalar irradiance meter would serve as the most efficient type of light detector. Unlike a vector irradiance sensor, which is directionally dependent, a scalar irradiance meter collects light evenly from all directions of the lower hemisphere within the water column.

The results of this research contribute valuable insights toward the development of an efficient optical sensor system that could play a crucial role in the early detection and management of oil spills and leaks along navigation routes, as well as near marine infrastructure.

The scientific significance of the obtained results primarily pertains to the southern Baltic Sea. Similar work can be carried out for other ocean regions, as well as for inland surface waters, but only as measurement data regarding the Inherent Optical Properties (IOPs) of these waters accumulates.

## Figures and Tables

**Figure 1 sensors-25-05267-f001:**
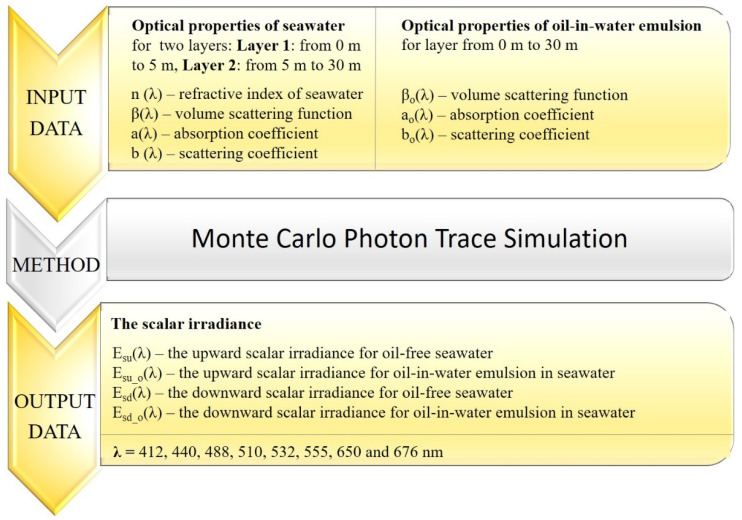
Block diagram of the model of the upward and downward scalar irradiance below the sea surface.

**Figure 2 sensors-25-05267-f002:**
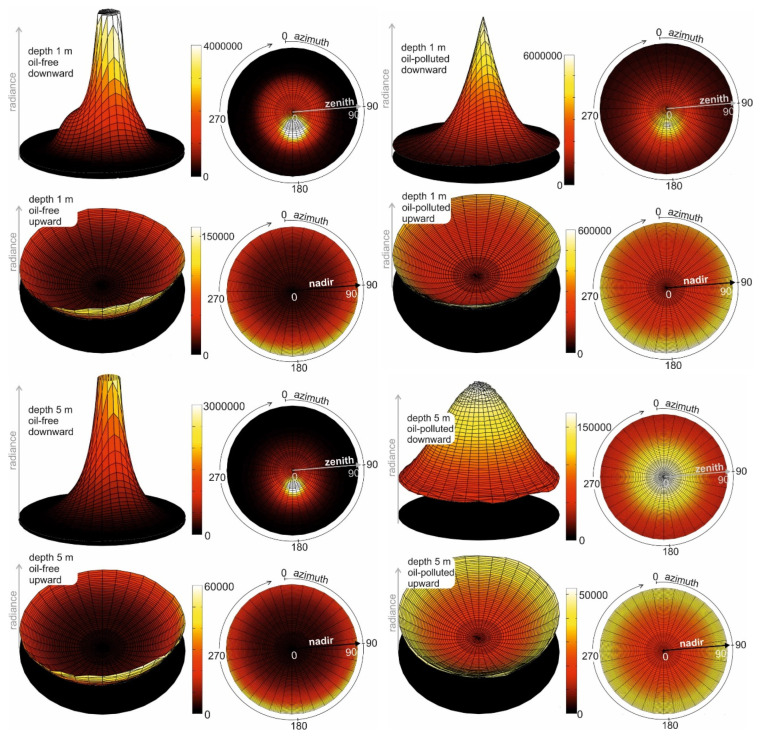
Directional distribution (as 3D and 2D graphs) of downward and upward radiance (in relative units) for oil-free seawater (on the **left**) and oil-polluted seawater (on the **right**) for 555 nm at chosen depths.

**Figure 3 sensors-25-05267-f003:**
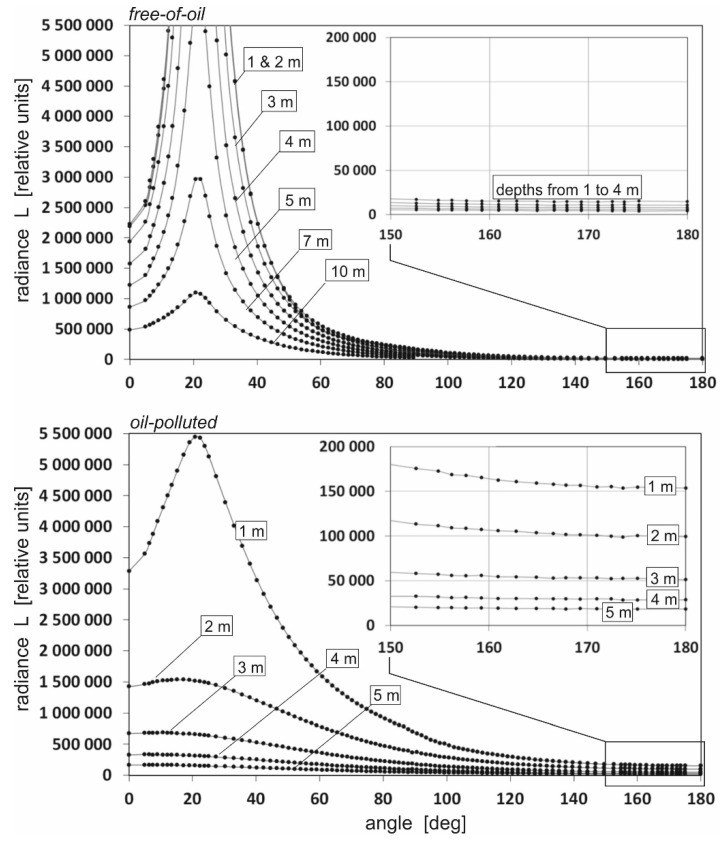
Example of the directional cross-section of downward radiance for oil-polluted seawater (on the **top**) and for free-of-oil seawater (at the **bottom**) for various depths (wavelength 555 nm, azimuth 0°).

**Figure 4 sensors-25-05267-f004:**
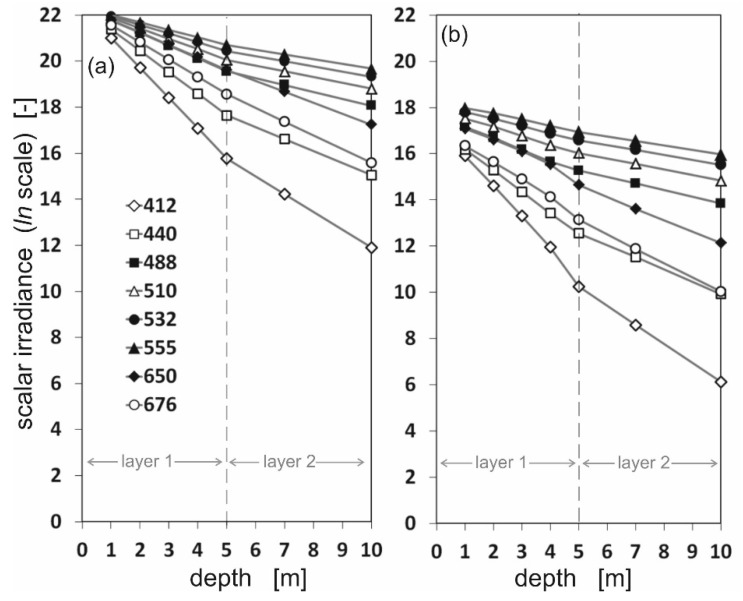
The values of downward (**a**) and upward (**b**) scalar irradiance for eight chosen wavelengths under the sea surface at seven depths: 1 m, 2 m, 3 m, 4 m, 5 m, 7 m, and 10 m, for oil-free seawater.

**Figure 5 sensors-25-05267-f005:**
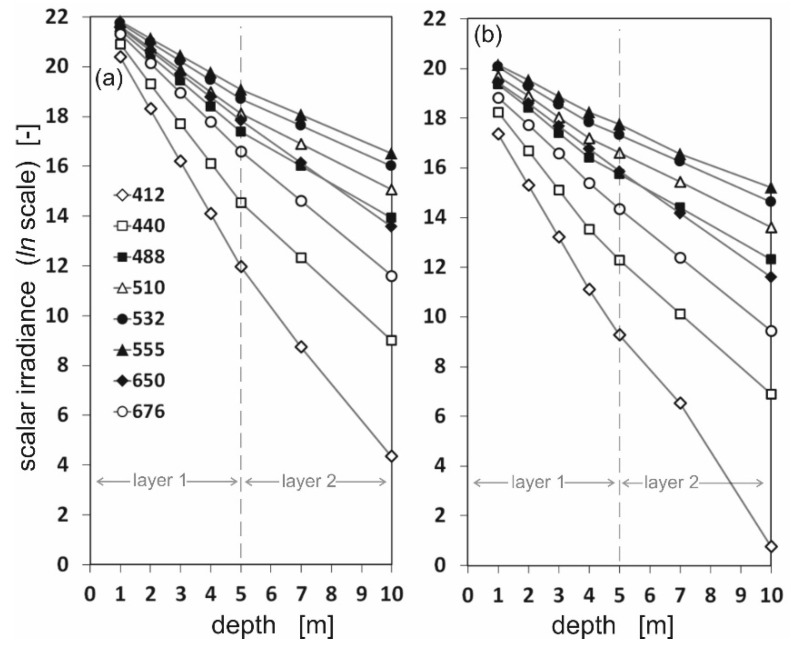
The values of downward (**a**) and upward (**b**) scalar irradiance for eight chosen wavelengths under the sea surface at seven depths: 1 m, 2 m, 3 m, 4 m, 5 m, 7 m, and 10 m, for oil-polluted seawater.

**Figure 6 sensors-25-05267-f006:**
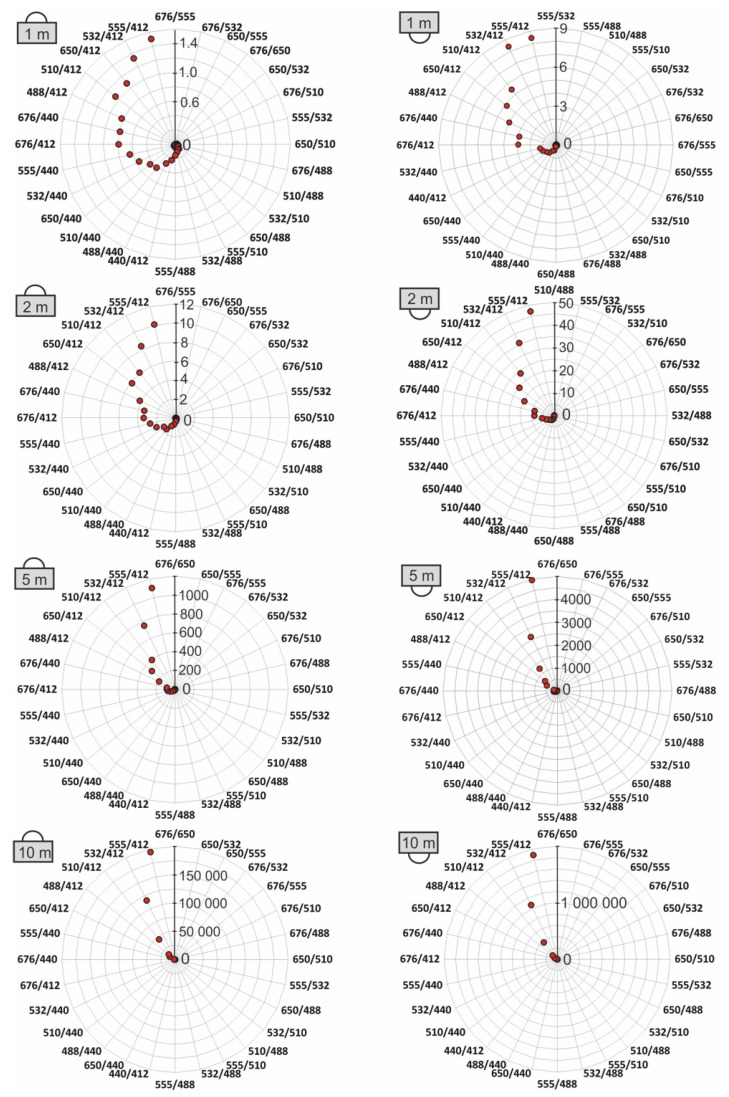
The differences between the values of the spectral index for oil-polluted seawater and oil-free seawater for downward scalar irradiance readings (irradiance meter directed upward) (**left**). The differences between the values of the spectral index for oil-polluted seawater and oil-free seawater for upward scalar irradiance readings (irradiance meter directed downward) (**right**). The scales on the radial axes were selected for each case separately in order to best visualize the obtained results. Values of differences for every wavelength combination in Table 3 and Table 4 are shown.

**Figure 7 sensors-25-05267-f007:**
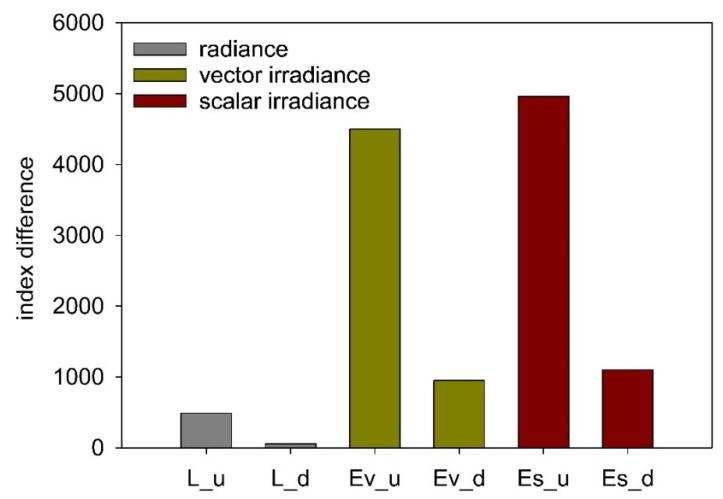
The differences between the values of the spectral index for oil-polluted seawater and oil-free seawater for upward radiance (L_u) [16], downward radiance (L_d) [17], upward vector irradiance (Ev_u) [18], downward vector irradiance (Ev_d) [18], upward scalar irradiance (Es_u), and downward scalar irradiance (Es_d) readings at a depth of 5 m.

**Table 1 sensors-25-05267-t001:** The values of absorption (*a* (λ)) (sum of absorption by dissolved substances, suspended particles, and water molecules) and scattering (*b* (λ)) (sum of scattering by suspended particles and water molecules) coefficients for southern Baltic Sea waters for the eight wavelengths, based on [21].

Wavelength (*λ*) [nm]	Layer 1 (0–5 m)		Layer 2 (5–30 m)	
	Absorption Coefficient (*a*) [m^−1^]	Scattering Coefficient (*b*) [m^−1^]	Absorption Coefficient (*a*) [m^−1^]	Scattering Coefficient (*b*) [m^−1^]
412	0.596	0.63	0.536	0.39
440	0.398	0.60	0.348	0.37
488	0.218	0.60	0.178	0.37
510	0.188	0.60	0.158	0.37
532	0.163	0.60	0.143	0.37
555	0.149	0.59	0.139	0.37
650	0.391	0.54	0.381	0.34
676	0.517	0.51	0.497	0.32

**Table 2 sensors-25-05267-t002:** The values of absorption and scattering coefficients of oil-in-water emulsions (*a_o_* (λ) and *b_o_* (λ)) with a concentration of 10 ppm for the eight wavelengths [18].

Wavelength (*λ*) [nm]	Absorption Coefficient (*a_o_*) [m^−1^]	Scattering Coefficient (*b_o_*) [m^−1^]
412	0.299	7.81
440	0.114	7.97
488	0.052	7.98
510	0.042	7.95
532	0.029	7.91
555	0.029	7.87
650	0.0125	7.60
676	0.0087	7.48

**Table 3 sensors-25-05267-t003:** The differences between the values of the spectral index for oil-polluted seawater and oil-free seawater for downward scalar irradiance (irradiance meter directed upward) readings for chosen depths (*λ_i_* = 412 nm, 440 nm, 488 nm, 510 nm, 532 nm, 555 nm, 650 nm, and 676 nm).

*λ_i_*/*λ_i_*	1 m	*λ_i_*/*λ_i_*	2 m	*λ_i_*/*λ_i_*	5 m	*λ_i_*/*λ_i_*	10 m
555/412	1.502	555/412	10.094	555/412	1103.417	555/412	194,865.181
532/412	1.325	532/412	8.376	532/412	747.955	532/412	115,773.694
650/412	1.082	510/412	6.085	510/412	393.895	510/412	44,999.828
510/412	1.063	650/412	5.856	650/412	309.666	488/412	14,288.185
488/412	0.827	488/412	4.184	488/412	182.514	650/412	10,301.518
676/440	0.785	676/440	3.368	676/440	86.555	555/440	1736.063
676/412	0.785	676/412	3.368	676/412	86.555	676/440	1415.364
555/440	0.648	555/440	2.763	555/440	73.664	676/412	1415.364
532/440	0.555	532/440	2.233	532/440	48.990	532/440	1022.643
650/440	0.446	650/440	1.546	510/440	24.511	510/440	386.106
510/440	0.416	510/440	1.521	650/440	20.056	488/440	117.189
488/440	0.296	488/440	0.962	488/440	10.585	650/440	88.965
440/412	0.224	440/412	0.692	440/412	6.529	440/412	83.961
555/488	0.154	555/488	0.395	555/488	2.322	555/488	8.374
532/488	0.113	532/488	0.274	532/488	1.347	532/488	4.441
555/510	0.084	555/510	0.190	555/510	0.756	555/510	1.892
650/488	0.081	650/488	0.178	650/488	0.503	510/488	1.034
532/510	0.051	532/510	0.110	510/488	0.400	532/510	0.864
510/488	0.049	510/488	0.108	532/510	0.370	650/488	0.271
676/488	0.036	676/488	0.065	555/532	0.149	555/532	0.260
650/510	0.031	650/510	0.064	650/510	0.119	650/510	0.017
555/532	0.026	555/532	0.052	676/488	0.083	676/488	0.015
676/510	0.001	676/510	0.004	676/510	−0.002	676/510	−0.008
650/532	−0.010	650/532	−0.011	650/532	−0.024	676/555	−0.009
676/650	−0.026	676/532	−0.037	676/532	−0.032	676/532	−0.011
650/555	−0.029	650/555	−0.039	676/555	−0.034	650/555	−0.034
676/532	−0.031	676/650	−0.041	650/555	−0.051	650/532	−0.035
676/555	−0.044	676/555	−0.051	676/650	−0.057	676/650	−0.050

**Table 4 sensors-25-05267-t004:** The differences between the values of the spectral index for oil-polluted seawater and oil-free seawater for upward scalar irradiance readings (irradiance meter directed downward) for chosen depths (*λ_i_* = 412 nm, 440 nm, 488 nm, 510 nm, 532 nm, 555 nm, 650 nm, and 676 nm).

*λ_i_*/*λ_i_*	1 m	*λ_i_*/*λ_i_*	2 m	*λ_i_*/*λ_i_*	5 m	*λ_i_*/*λ_i_*	10 m
555/412	8.409	555/412	47.131	555/412	4962.354	555/412	1,892,457.173
532/412	8.356	532/412	35.628	532/412	2622.654	532/412	1,063,963.351
510/412	5.409	510/412	23.784	510/412	1245.869	510/412	378,443.283
650/412	4.806	650/412	19.636	650/412	659.915	488/412	104,146.424
488/412	3.941	488/412	14.620	488/412	503.482	650/412	50,914.571
676/440	2.859	676/440	8.780	555/440	154.984	676/440	5900.442
676/412	2.859	676/412	8.780	676/440	147.071	676/412	5900.442
532/440	1.228	555/440	5.418	676/412	147.071	555/440	3639.459
440/412	1.088	532/440	3.875	532/440	95.608	532/440	2015.354
650/440	0.882	650/440	2.835	510/440	42.350	510/440	680.630
555/440	0.766	510/440	2.419	650/440	27.178	440/412	426.386
510/440	0.521	440/412	2.125	488/440	16.106	488/440	175.706
488/440	0.437	488/440	1.476	440/412	11.002	650/440	99.796
650/488	0.154	650/488	0.262	555/488	2.207	555/488	9.607
676/488	0.149	555/488	0.202	532/488	1.142	532/488	4.738
532/488	0.130	676/488	0.162	555/510	0.668	555/510	1.883
650/510	0.125	650/510	0.159	650/488	0.592	510/488	0.906
532/510	0.125	555/510	0.119	532/510	0.301	532/510	0.813
676/510	0.114	676/510	0.099	510/488	0.243	650/488	0.298
650/555	0.084	650/532	0.095	650/510	0.221	555/532	0.221
676/555	0.075	532/488	0.085	676/488	0.132	650/510	0.065
676/650	0.070	650/555	0.063	555/532	0.116	676/488	0.033
676/532	0.059	676/532	0.062	650/532	0.087	650/532	0.013
650/532	0.044	676/650	0.053	676/510	0.049	676/510	0.007
555/510	−0.011	532/510	0.047	676/555	0.049	650/555	0.005
510/488	−0.035	676/555	0.044	676/532	0.020	676/532	0.001
555/488	−0.071	555/532	0.039	676/555	0.011	676/555	0.000
555/532	−0.112	510/488	0.005	676/650	0.000	676/650	−0.004

## Data Availability

The datasets used and analyzed during the current study are available from the corresponding author upon request.

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
