# Peer review of "Spectral and Geometrical Guidelines for Low-Concentration Oil-in-Seawater Emulsion Detection Based on Monte Carlo Modeling"

_sensors, 2025, doi:10.3390/s25175267_

Round 1

Reviewer 1 Report

Comments and Suggestions for Authors

21.07.2025

Review of the Article:

Spectral and geometrical guidelines for low concentration oil-in-seawater emulsion detecting based on Monte Carlo modelling, by Barbara Lednicka and Zbigniew Otremba in ‘SENSORS’.

The work highlights a significant issue regarding the need for rapid and simple detection of oil pollution, which results from the expansion of maritime infrastructure. Since the 1970s, there have been several hundred major oil spill incidents. Therefore, the authors aim to establish design assumptions for a sensor that detects oil dispersed in seawater (oil-in-water emulsions). The approach relies on analysing changes in the underwater solar radiance (L) field caused by oil droplets in the water column. The paper proposes an interdisciplinary approach, combining modelling (Monte Carlo simulation) and empirically measured data, to develop tools for monitoring human impact (in this case, oil pollution) on the marine environment. The core principle of the analysis involves simulating the paths of numerous virtual solar photons within a seawater model defined by spatially distributed Inherent Optical Properties (IOPs): the in situ data of scattering and absorption coefficients (listed in Tab. 1) for the southern Baltic Sea, along with Petrobaltic oil samples (this kind of oil is extracted from the seabed in this area) (shown in Tab. 2). 

The authors focus on digitally modelling an underwater optical sensor specifically designed to measure the presence and concentration of oil pollutants throughout the water column, not just at the surface. Therefore, the virtual receivers were ‘positioned’ at different depths. In the optical model of a 30 m oil-polluted water layer, an oil droplet concentration of 10 ppm was assumed. As a result, eight wavelengths covering nearly the entire visible spectral range, from 412 nm to 676 nm, were selected for the study.

As a result, one has the vertical distribution of downward radiance and upward radiance for both oil-free and oil-polluted seawater, as well as the directional radiance distributions for the chosen wavelengths for chosen depths (1 m, 2 m, 5 m, and 10 m). The angular radiance distributions obtained by simulating the fate of a large number of virtual photons (one hundred million) in the oil-free and oil-contaminated sea differ.

Next, it was confirmed that the spectral index of 555/412 is the most suitable combination of two wavelengths for underwater detection of oil-in-water emulsions. Furthermore, based on a comparison with previous research, the presented results provided strong evidence that scalar irradiance should be preferred over radiance or vector irradiance sensors for detecting oil-in-water emulsions.

The authors also highlight the limitations of this method. Radiance in the water column depends on several factors, primarily the spectral and angular distribution of solar light incident on the sea surface, the condition of the sea surface, depth, and the Inherent Optical Properties (IOPs) of seawater, which include the optical properties of both solid and dissolved components in water. Variations in the radiance field in the sea (with oil emulsion) can be caused not only by the oil emulsion but also by other elements such as bubbles, resuspended bottom sediment, and phytoplankton blooms.

This finding is a highly advantageous result from a technical perspective, as a downward-facing sensor is less likely to accumulate various types of contaminants, such as organic and mineral deposits.

The results of this research provide valuable insights for developing efficient optical sensor systems that could play a crucial role in the early detection and management of oil spills and leaks near marine infrastructure.

I have one question: it was stated that shorter wavelengths are absorbed by Coloured Dissolved Organic Matter (CDOM), while water molecules absorb longer wavelengths. Why didn’t you attempt to use an index with orange colour (555 nm) and red (676 nm), such as 555/676? Perhaps the result would be similar to this with 555/412. Additionally, could you provide a physical explanation for why the index 555/412 is so effective at detecting oil-in-seawater emulsion?

In my opinion, all the steps of the research are very well explained.

The figures are legible. The figure captions are adequate to the content of the graphs and contain all essential details. However, an explanation of Fig. 2. could be slightly improved in the text.

The language is clear, and the text is well-written.

There are only three minor typos.

  1. Instead of Colored Dissolved Organic Matter (CDOM), it should be Chromophoric Dissolved Organic Matter (CDOM).
  2. Page 10. In a sentence: ‘Based on a comparison of the present and previous studies, the present results provide strong evidence that a solar irradiance should be chosen instead of a radiance or vector irradiance sensor for the detection of oil-in-water emulsions (Fig. 7).’

Instead of ‘solar’, it should be ‘scalar’.

  1. Page 11. In Conclusions.

First two paragraphs: ‘The research focused on identifying optimal structural and functional parameters for an optical remote system capable of detecting environmentally hazardous oil substances - in this case oil dispersed in seawater (oil-in-water emulsion). Through numerical simu-lation, the study confirms that employing a measurement system based on the detection of solar light intensity at two specific wavelengths approximately 555 nm (green) and 412 nm (blue) is optimal. The findings presented in this work suggest that a scalar irradiance meter would serve as the most efficient type of light detector. Unlike vector irradiance sensors, which are directionally dependent, a scalar irradiance meter collects light evenly from all directions of down hemisphere within the water column.

The results of this research contribute valid insights toward the development of efficient optical sensor system that could play a crucial role in the early detection and management of oil spills and leaks near marine infrastructure.’ – should be removed. The following sentences repeat it almost identically.

Author Response

Reviewer 1

Thank you for your comment and suggestions.

Comments 1:

I have one Comments: it was stated that shorter wavelengths are absorbed by Coloured Dissolved Organic Matter (CDOM), while water molecules absorb longer wavelengths. Why didn’t you attempt to use an index with orange colour (555 nm) and red (676 nm), such as 555/676? Perhaps the result would be similar to this with 555/412. Additionally, could you provide a physical explanation for why the index 555/412 is so effective at detecting oil-in-seawater emulsion?

Response on 1:

As shown in Tables 3 and 4, the most efficient index for measurement would be 555/412. However, the 532/412 and 650/412 indices also show sensitivity to the presence of dispersed oil. Therefore this isn't about the exact values of the 555 nm and 412 nm wavelengths, but values close to them. In our study, we analyzed a large number of possible indices (28 cases in a wide range of the visible spectrum), with the choice of individual wavelengths resulting from the availability of IOPs for such wavelengths in the literature.  The physical explanation for the optimal index consisting of green and blue wavelengths stems from the interaction of light absorption by oil, light scattering by oil droplets, the angular scattering function of oil droplets, light absorption by seawater components, light scattering by suspensions, and the angular scattering function of suspension components. Monte Carlo photon simulations take all of these factors into account.

Comments 2:

The figures are legible. The figure captions are adequate to the content of the graphs and contain all essential details. However, an explanation of Fig. 2. could be slightly improved in the text.

Response on 2:

We took the suggestion of yours, for which we thank you, and we overwrote the sentences: „Figure 2 presents the results of the staged simulations, specifically the angular distributions of downward and upward radiance for both oil-free and oil-contaminated seawater at a selected wavelength (555 nm) and depths (1 m and 5 m). The radiance distributions were obtained by simulating the trajectories of a large number of virtual photons (one hundred million) within both clean and oil-polluted marine environments. Significant differences are observed between the two cases. Based on these distributions, both scalar and vector irradiance values were calculated.”

Comments 3:

Instead of Colored Dissolved Organic Matter (CDOM), it should be Chromophoric Dissolved Organic Matter (CDOM).

Response on 3:

The term "Colored Dissolved Organic Matter" (CDOM) is indeed often used interchangeably with Chromophoric Dissolved Organic Matter (CDOM). However, we agree that "Chromophoric" is more scientifically, we decided to change "Colored" to "Chromophoric" in our article, as suggested.

Comments 4:

Page 10. In a sentence: ‘Based on a comparison of the present and previous studies, the present results provide strong evidence that a solar irradiance should be chosen instead of a radiance or vector irradiance sensor for the detection of oil-in-water emulsions (Fig. 7).’ Instead of ‘solar’, it should be ‘scalar’.

Response on 4:

We changed ‘solar‘ to ‘scalar’ in this sentence.

Comments 5:

First two paragraphs: ‘The research focused on identifying optimal structural and functional parameters for an optical remote system capable of detecting environmentally hazardous oil substances - in this case oil dispersed in seawater (oil-in-water emulsion). Through numerical simu-lation, the study confirms that employing a measurement system based on the detection of solar light intensity at two specific wavelengths approximately 555 nm (green) and 412 nm (blue) is optimal. The findings presented in this work suggest that a scalar irradiance meter would serve as the most efficient type of light detector. Unlike vector irradiance sensors, which are directionally dependent, a scalar irradiance meter collects light evenly from all directions of down hemisphere within the water column.

The results of this research contribute valid insights toward the development of efficient optical sensor system that could play a crucial role in the early detection and management of oil spills and leaks near marine infrastructure.’ – should be removed. The following sentences repeat it almost identically.

Response on 5:

These first two paragraphs have been removed.

Reviewer 2 Report

Comments and Suggestions for Authors

The article is devoted to determining the degree of contamination of seawater with oil in specific conditions (water transparency, reflection and absorption coefficients, backscattering, etc.). This is the main disadvantage of the article. The article does not show how their results depend on these parameters, which can vary not only from sea to sea, but also within the same sea for coastal regions. Therefore, the article cannot be recommended for publication in this form. It is necessary to repeat our research in the article for different natural conditions (transparency, reflection and absorption coefficients, backscattering, the presence of primary biomass, aerosol, etc.). In addition, for all seas, "determining the intensity of sunlight at two specific wavelengths, approximately 555 nm (green) and 412 nm (blue), is it optimal"? I think that's not the case. It is necessary to prove this using diametrically opposed examples.

Author Response

Reviewer 2

Thank you for your comment and suggestions.

Comments 1:

The article is devoted to determining the degree of contamination of seawater with oil in specific conditions (water transparency, reflection and absorption coefficients, backscattering, etc.). This is the main disadvantage of the article. The article does not show how their results depend on these parameters, which can vary not only from sea to sea, but also within the same sea for coastal regions. Therefore, the article cannot be recommended for publication in this form. It is necessary to repeat our research in the article for different natural conditions (transparency, reflection and absorption coefficients, backscattering, the presence of primary biomass, aerosol, etc.). In addition, for all seas, "determining the intensity of sunlight at two specific wavelengths, approximately 555 nm (green) and 412 nm (blue), is it optimal"? I think that's not the case. It is necessary to prove this using diametrically opposed examples.

Response on 1:

Thank you for sharing your thoughts, which will contribute in planning of future research related to the applicability of dispersed oil detection in waters with various optical properties, under different weather conditions, and in relation to possible types of oil and its concentration in water. But if we wanted to response all your comments about other feasible alternatives: transparency, reflection and absorption coefficients, backscattering, the presence of primary biomass, aerosol, etc., we would have to write many separate papers. It is obvious that this model can include all of the above. However, we wanted to model the upward and downward scalar irradiance detected by a virtual underwater sensor for specific selected input parameters for comparison with already published results of the upward and downward vector irradiance and the upward and downward radiance, carried out with precisely these initial conditions. The observations presented in this paper, taking into account the conditions of the Southern Baltic both in terms of the optical nature of the waters and the oil extracted in this area, open the possibility of building a model with a wide sphere of universality, considering the nature of the other parts of the ocean.

To further highlight the benefits of our article, we have added the following text to the discussion and conclusions sections.

For discussion:

This study builds upon previous research in the field of marine remote sensing by incorporating an additional and often overlooked variable: the total amount of light energy available in all directions at a given point underwater, also known as scalar irradiance. While earlier studies primarily focused on directional light measurements such as radiance or vector irradiance this work emphasizes the importance of capturing the omnidirectional nature of light as it interacts with the aquatic environment.

Importantly, the findings demonstrate that oil pollution can be detected even when the oil is present in the form of highly diluted emulsions. This highlights a significant advancement in detection capability, as it opens the door to identifying subtle or early-stage pollution events that might otherwise go unnoticed until they manifest more severely.

The omnidirectional sensitivity of the detector is particularly beneficial in deeper waters, where light levels are naturally reduced and directional measurements using radiance meters become less reliable. The ability to detect subtle differences in spectral indices at greater depths extends the range of detection instruments based on the scalar irradiance meter principle, making them more suitable for long-term environmental monitoring in marine environments. For example, while radiance meters become virtually useless at low light levels, scalar irradiance sensors can still operate effectively, providing more consistent data acquisition in difficult lighting conditions.

Another key observation is the depth-dependent behaviour of detector sensitivity. The results show a notable increase in the sensitivity of the spectral index to oil presence as the sensor is submerged deeper. This is likely due to the cumulative optical effects that oil has on light as it travels through water. However, this increased sensitivity is limited by a concurrent decrease in the total available light energy at greater depths, which may impose practical limits on the sensor’s effectiveness if ambient light falls below a functional threshold.

The findings suggest that a scalar irradiance meter would serve as the most efficient type of light detector for this purpose. Unlike vector irradiance sensors, which are directionally dependent, a scalar irradiance meter collects light from all upward directions within the water column, making it particularly suited for capturing diffuse light. This is critical in marine environments, where suspended matter and varying light conditions can significantly affect detection accuracy.

The simulations and analysis were conducted with a model of seawater exhibiting relatively high turbidity, characteristic of the Baltic Sea. In such environments, light attenuation is more pronounced due to the presence of suspended particulate matter, CDOM, and other optically active substances. Despite these challenges, the results showed that it should still be possible to detect oil pollution by optimizing the detection parameters, such as wavelength selection and sensor orientation.

It is worth noting that in marine areas with higher water transparency, such as open ocean waters or tropical seas, the detection of oil dispersed in the water column is expected to be more effective. This is because lower turbidity allows light to travel further, enhancing the detector's ability to distinguish oil from the surrounding water. Therefore, while the system modeled here is optimized for the relatively challenging optical conditions of the Baltic, it holds even greater promise for application in clearer waters.’

For conclusions:

‘The scientific significance of the obtained results primarily pertains to the southern Baltic Sea. Similar work can be carried out for other ocean regions, as well as for inland surface waters, but only as measurement data regarding the Inherent Optical Properties (IOPs) of these waters accumulates.’

Reviewer 3 Report

Comments and Suggestions for Authors

In this paper, the authors have tried to display the result of the search for design assumptions for a sensor to detect oil dispersed in the sea waters based on analysing changes in the underwater solar radiance field caused by the presence of oil droplets in the water column. This work is very interesting. However, the paper lacks the presentation of measured data and also the display of characteristics of key wavelengths such as ultraviolet and near-infrared. The main comments are as follows:

  • Many variable symbols are not italicized.
  • What is the evidence that the index 555/412 is also the most favourable index?
  • Are downward and upward vector irradiance easy to measure? Are they easy to obtain?
  • Figure1: What is the basis for the selection of wavelengths?
  • Figure1: 4888 may be a typo.
  • Conclusions: It is suggested that the authors enhance the practicality of the research content in this paper. For example, by how much can the accuracy be improved? Is the amount of data required reduced? Are the required variables easier to measure? In short, the paper should better highlight its contributions to this field.

Author Response

Reviewer 3

Thank you for your comment and suggestions.

Comments 1 and 5:

(1) However, the paper lacks the presentation of measured data and also the display of characteristics of key wavelengths such as ultraviolet and near-infrared.

(5) Figure1: What is the basis for the selection of wavelengths?

Response on 1 and 5:

We adopted wavelengths (from the range of water-penetrating visible light) for which inherent optical parameters (IOPs) are available in the literature for the waters of the southern Baltic Sea. Most ocean radiometers operate within the visible spectrum (approximately 400–700 nm), also known as the optical window, where sunlight penetrates water most efficiently. This range is optimal for detecting variations in water-leaving radiance influenced by phytoplankton, dissolved matter, and particulates (for example: 443 nm – strong chlorophyll absorption, 555 nm – minimal absorption, 670–685 nm - chlorophyll fluorescence peak and absorption feature). Because it is known that different constituents absorb and scatter light differently at various wavelengths so we can used that to distinguish of water constituents (for example: CDOM – absorbs strongly in the blue and UV (~350–450 nm), suspended sediments – increase scattering, particularly in the red and NIR (~600–900 nm), Oil pollution – alters radiance mostly in the blue-green region). Moreover bands in radiometers can be chosen to avoid atmospheric absorption features (but this issue does not concern us), particularly from oxygen and water vapor. Atmospheric correction requires bands in the near-infrared (NIR) (~765–865 nm) and shortwave infrared (SWIR) (~1240, 1640, 2130 nm) for land–water distinction and aerosol estimation over water.

Comments 2:

Many variable symbols are not italicized.

Response on 2:

We took the suggestion of yours, for which we thank you, and we used italic for all symbols in article.

Comments 3:

What is the evidence that the index 555/412 is also the most favourable index?

Response on 3:

Figure 6 and Table 3 and 4 show that the highest value of the index for oil-free water and dispersed oil is recorded for the wavelength combination 555/412. An it confirmed that the spectral index of 555/412 is the most favourable combination of two wavelengths for the underwater detection of oil-in-water emulsions.

Comments 4:

Are downward and upward vector irradiance easy to measure? Are they easy to obtain?

Response on 4:

The device for which we have prepared the basic design assumptions would be simpler than some operational multispectral devices (e.g., Satlantic radiometers), because it would be designed to measure only two wavelengths. Underwater data transmission would likely pose a greater challenge if the sensor were installed underwater or on an unmanned underwater inspection vehicle.

Comments 6:

Figure1: 4888 may be a typo.

Response on 6:

We corrected the typo in the Figure 1. Now there is 488.

Comments 7:

Conclusions: It is suggested that the authors enhance the practicality of the research content in this paper. For example, by how much can the accuracy be improved? Is the amount of data required reduced? Are the required variables easier to measure? In short, the paper should better highlight its contributions to this field.

Response on 7:

The required variables are: the values of absorption (a(λ)) (sum of absorption by dissolved substances, suspended particles and water molecules) and scattering (b(λ)) (sum of scattering by suspended particles and water molecules) coefficients for southern Baltic Sea waters for the eight wavelengths, based on Sagan [21]. This parameters can be measured by an underwater absorption and attenuation meter ac-9 (WET Labs, Philomath, OR, USA), which measure these values for nine wavelengths (412 nm, 440 nm, 488 nm, 510 nm, 532 nm, 555 nm, 650 nm, 676 nm and 715 nm). The scattering coefficient by suspended constituents of seawater (bp) can be determined based on the subtraction of these two quantities. The ac-9 requires corrections for temperature and salinity effects. These corrections are crucial because the optical properties of water change with temperature and salinity.

Our study addresses the pressing need for fast and straightforward detection methods for oil contamination, a growing concern linked to the expanding maritime infrastructure.  The methodology is based on analysing alterations in the underwater solar radiance field induced by the presence of oil droplets in the water column. An interdisciplinary strategy is adopted, integrating numerical modelling - specifically Monte Carlo simulations with empirical data collection to create tools for assessing anthropogenic impacts on marine ecosystems, particularly oil pollution. The central analytical approach involves simulating the trajectories of numerous virtual solar photons through a seawater model characterized by spatially varying Inherent Optical Properties (IOPs). These IOPs include site-specific absorption and scattering coefficients for the southern Baltic Sea, as well as optical properties of Petrobaltic crude oil extracted from this region. The primary objective is the digital development of a submerged optical sensor tailored to detect oil contamination not only at the surface but throughout the entire water column. To that end, virtual detectors were positioned at varying depths. The simulations modelled a 30-meter-thick oil-polluted water layer, assuming an oil droplet concentration of 10 ppm. Eight wavelengths, spanning most of the visible light spectrum (from 412 nm to 676 nm), were selected for the simulation. As a result, vertical profiles of both downward and upward radiance were obtained for oil-free and oil-contaminated seawater. In addition, angular radiance distributions were calculated for selected depths (1 m, 2 m, 5 m, and 10 m) at the specified wavelengths. The radiance patterns reveal distinct differences between clean and polluted conditions, based on the propagation of one hundred million virtual photons in each scenario. Further analysis identified the spectral index 555/412 as the most effective wavelength pair for detecting dispersed oil in the water column. The results also demonstrate, in agreement with earlier studies, that scalar irradiance measurements are more reliable than those based on radiance or vector irradiance for detecting oil-in-water emulsions. However, we acknowledge the limitations of this method. The radiance field within the ocean is influenced by numerous variables, including the spectral and angular distribution of incoming solar radiation, surface conditions, depth, and the IOPs of the water. These optical properties encompass both particulate and dissolved matter. Consequently, changes in the radiance field attributed to oil emulsions may also result from other factors such as gas bubbles, resuspended sediments, or phytoplankton blooms. From a technical standpoint, one notable advantage of the proposed approach is that a downward-facing sensor configuration is less susceptible to fouling from organic or mineral deposits. Overall, our findings offer valuable guidance for designing advanced optical sensing technologies, which could significantly enhance the early detection and mitigation of oil spills and leaks in areas near marine infrastructure.

Round 2

Reviewer 2 Report

Comments and Suggestions for Authors

The article can be published.

Reviewer 3 Report

Comments and Suggestions for Authors

The author has made quite comprehensive revisions, and I agree that the paper be accepted for publication.